# TRANSFORMERS VERSUS LSTMS FOR ELECTRONIC TRADING

## ABSTRACT

The rapid advancement of artificial intelligence has seen widespread application of long short-term memory (LSTM), a type of recurrent neural network (RNN), in time series forecasting. Despite the success of Transformers in natural language processing (NLP), which prompted interest in their efficacy for time series prediction, their application in financial time series forecasting is less explored compared to the dominant LSTM models. This study investigates whether Transformer-based models can outperform LSTMs in financial time series forecasting. It involves a comparative analysis of various LSTM-based and Transformer-based models on multiple financial prediction tasks using high-frequency limit order book data. A novel LSTM-based model named DLSTM is introduced alongside a newly designed Transformer-based model tailored for financial predictions. The findings indicate that Transformer-based models exhibit only a marginal advantage in predicting absolute price sequences, whereas LSTM-based models demonstrate superior and more consistent performance in predicting differential sequences such as price differences and movements.

## 1 INTRODUCTION

LSTM has been proven successful in the application of sequential data. Like LSTM, the Transformer Vaswani et al. (2017) is also used to handle the sequential data. Compared to LSTM, the Transformer does not need to handle the sequence data in order, which instead confers the meaning of the sequence by the Self-attention mechanism.

Since 2017, the Transformer has been increasingly used for Natural Language Processing (NLP) problems. It produces more impressive results than RNN, such as machine translation Lakew et al. (2018) and speech applications Karita et al. (2019), replacing RNN models such as LSTM in NLP tasks. Recently, a surge of Transformer-based solutions for less explored long time series forecasting problem has appeared Wen et al. (2022). However, as for the financial time series prediction, LSTM remains the dominant architecture.

Investigating whether Transformer-based methods are suitable for financial time series forecasting is the central focus of this paper, which compares the efficacy of Transformer and LSTM-based approaches using LOB data from Binance Exchange across various financial prediction tasks. These tasks include mid-price prediction, mid-price difference prediction, and mid-price movement prediction. In the first two tasks, the study assesses existing Transformer and LSTM models; for mid-price prediction, Transformer methods show a $10\% - 25\%$ lower prediction error than LSTM methods, although the results are not sufficiently reliable for trading. Conversely, LSTM models excel in mid-price difference prediction, achieving an out-of-sample $R^2$ of approximately $11.5\%$. The paper's most notable contribution is the development of a new LSTM-based model, DLSTM, specifically designed for mid-price movement prediction by integrating LSTM with a time series decomposition approach. This model significantly outperforms previous methods, with accuracy ranging from $63.73\%$ to $73.31\%$, demonstrating robust profitability in simulated trading scenarios. Moreover, the architecture of existing Transformer-based methods has been modified to better suit the demands of movement prediction tasks.

## 2 LSTM in Time Series Prediction

LSTM, introduced by Hochreiter et al. Hochreiter and Schmidhuber (1997), has become a cornerstone for time series prediction, especially in handling long-term dependencies that are beyond the reach of traditional Recurrent Neural Networks (RNN). RNN often struggles with issues like exploding or vanishing gradients, which impede the learning of long-range dependencies Rumelhart et al. (1986); Goodfellow et al. (2016). LSTMs mitigate these problems through a series of gating mechanism that regulates information flow, thus maintaining model stability over extended sequences Gers et al. (1999).

In the financial sector, LSTMs have proven particularly effective, being widely applied in predicting stock prices using Open-High-Low-Close (OHLC) data and other financial indices Roondiwala et al. (2017); Cao et al. (2019); Bao et al. (2017); Selvin et al. (2017); Fischer and Krauss (2018). Notably, models such as Bidirectional LSTM (BiLSTM) and hybrids of LSTM with Convolutional Neural Networks (CNN) have further enhanced prediction accuracy Siami-Namini et al. (2019); Zhang et al. (2019).

Zhang et al. expanded LSTM's capabilities by developing the DeepLOB architecture, which incorporates convolutional blocks for feature extraction, an Inception module for decomposing inputs, and an LSTM layer to capture temporal patterns Zhang et al. (2019). This model excels in complex financial environments, particularly when analyzing high-frequency data from Limit Order Books (LOB). Further adaptations include DeepLOB-Seq2Seq and DeepLOB-Attention models, which integrate Seq2Seq and attention mechanisms, respectively, to improve multi-horizon and long-term predictions Zhang and Zohren (2021). These enhancements allow the models to handle more complex prediction tasks, achieving better performance by adapting the encoder-decoder framework for dynamic financial markets.

Such innovations demonstrate LSTM's adaptability and its continuous evolution to meet the specific demands of financial time series prediction, showcasing the model's robustness and reliability in capturing and analyzing intricate market dynamics.

## 3 Transformer in Time Series Prediction

The Transformer, originally impactful in natural language processing (NLP) Brown et al. (2020), has been adapted to tackle the unique challenges of time series prediction, particularly in financial contexts. According to Vaswani et al. Vaswani et al. (2017), the Transformer architecture employs a self-attention mechanism that efficiently processes long sequences without encountering the vanishing gradient problems typical of RNNs. This capability is particularly beneficial in financial markets characterized by long input sequences.

In the financial domain, the deployment of Transformer models is on the rise, with applications in predicting stock prices using Temporal Fusion Transformers Hu (2021) and in forecasting cryptocurrency values, showing notable advantages over LSTMs Sridhar and Sanagavarapu (2021). Innovative uses also include combining Transformers with BERT for sentiment analysis, followed by Generative Adversarial Networks (GANs) for stock price prediction Sonkiya et al. (2021).

To address the high computational demands of traditional self-attention, which scales quadratically with sequence length, new Transformer models like LogTrans Li et al. (2019), Reformer Kitaev et al. (2020), Informer Zhou et al. (2020), Autoformer Wu et al. (2021), Pyraformer Liu et al. (2022), and FEDformer Zhou et al. (2022) have been introduced. These models reduce complexity through innovations including convolutional self-attention, reversible connections, and ProbSparse mechanisms, enhancing efficiency in processing long sequences. They also incorporate advanced decomposition methods and frequency domain transformations, significantly improving forecasting accuracy and efficiency. Originally validated on datasets like electricity consumption and solar energy, these optimized Transformers show great potential for financial time series forecasting, surpassing traditional LSTM models in handling complex dependencies and long data sequences Wen et al. (2022).

## 4 FINANCIAL TIME SERIES PREDICTION TASKS FORMULATION

This study compares LSTM-based and Transformer-based methods among three financial prediction tasks based on LOB data. Three tasks are listed below:

### 4.1 TASK 1: LOB MID-PRICE PREDICTION

The first task is to predict the LOB Mid-Price Prediction, which is to compare the ability to predict absolute price values similar to non-financial datasets in previous works Li et al. (2019); Zhou et al. (2020); Wu et al. (2021); Zhou et al. (2022); Liu et al. (2022). The definition of time series prediction is given below and shown in Figure 1:

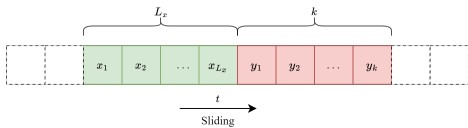

Figure 1: The illustration of time series prediction.

First, define a sliding window size $L_x$ for the past data. The input data at each time step $t$ is defined as:

$$X_t = \{x_1, x_2, \ldots, x_{L_x}\}_t \tag{1}$$

Then define a prediction window size $k$, where the goal is to predict the information in future $L_x + k$ steps. It will be the single-step prediction when $k = 1$ and be multi-horizon prediction when $k > 1$. Then the output at time step t is defined as:

$$Y_t = \{y_1, y_2, \ldots, y_k\}_t \tag{2}$$

The next step is to define the $x_t$ and $y_t$ in the input and output for mid-price prediction. Assume the market depth is 10. For a limit bid order at time t, the bid price is denoted as $p_{i,t}^{bid}$ and the volume is $v_{i,t}^{bid}$, where $i$ is the market depth. Same for the limit ask order, ask price is $p_{i,t}^{ask}$ and volume is $v_{i,t}^{ask}$. Then the LOB data at time t is defined as:

$$x_t = \left[p_{i,t}^{ask}, v_{i,t}^{ask}, p_{i,t}^{bid}, v_{i,t}^{bid}\right]_{i=1}^{n=10} \in R^{40} \tag{3}$$

The past mid-price will be added to LOB data as input, and the mid-price is represented as:

$$p_t^{\mathrm{mid}} = \frac{p_{1,t}^{ask} + p_{1,t}^{bid}}{2} \tag{4}$$

Finally, the $x_t$ will be:

$$x_t = \left[p_{i,t}^{ask}, v_{i,t}^{ask}, p_{i,t}^{bid}, v_{i,t}^{bid}, p_t^{\mathrm{mid}}\right]_{i=1}^{n=10} \in R^{41} \tag{5}$$

The target is to predict the future mid-price, so $y_t = p_t^{\mathrm{mid}}$.

### 4.2 TASK 2: LOB MID-PRICE DIFFERENCE PREDICTION

The second task is to predict the mid-price change, which is the the difference of two mid-prices in different time step. Trading strategies can be designed if the price change becomes negative or positive. The input of this task is the same as the mid-price prediction, as described in Equation 3. The target is to regress the future difference between current mid-price $p_t^{\mathrm{mid}}$ and the future mid-price $p_{t+\tau}^{\mathrm{mid}}$:

$$d_{t+\tau} = p_{t+\tau}^{\mathrm{mid}} - p_t^{\mathrm{mid}} \tag{6}$$

Like the mid-price prediction, a prediction window size is defined as $k$, then the output of this task in each timestamp $t$ is represented as:

$$Y_t = \{d_{t+1}, d_{t+2}, \ldots, d_{t+k}\}_t \tag{7}$$

## 4.3 TASK 3: LOB MID-PRICE MOVEMENT PREDICTION

To train a model to predict mid-price movement, the first step is to create price movement labels for each timestamp. This study follows the smoothing labelling method from Tsantekidis et al. Tsantekidis et al. (2017) and Zhang et al. Zhang et al. (2019): Use $m^-$ to represent the average of the last $k$ mid-price and $m^+$ to represent the average of the next $k$ mid-price:

$$m^-(t) = \frac{1}{k}\sum_{i=0}^{k} p_{t-k}^{mid} \tag{8}$$

$$m^+(t) = \frac{1}{k}\sum_{i=1}^{k} p_{t+k}^{mid} \tag{9}$$

$k$ is set to $20, 30, 50, 100$ in this study following previous work of Zhang et al. Zhang et al. (2019). And then, define a percentage change $l_t$ to decide the price change direction.

$$l_t = \frac{m^+(t) - m^-(t)}{m^-(t)} \tag{10}$$

The label is dependent on the value of $l_t$. A threshold $\delta$ is set to decide the corresponding label. There are three labels for the price movement:

$$\text{label} = \begin{cases} 0(\text{ fall }), & \text{when } l_t > \delta \\ 1(\text{ stationary }), & \text{when } -\delta \le l_t \le \delta \\ 2(\text{ rise }), & \text{when } l_t < -\delta \end{cases} \tag{11}$$

Assume there is an input in Equation 3 at timestamp $t$, predicting mid-price movement is a one-step ahead prediction, which is to predict the mid-price movement in timestamp $t + 1$.

## 5 EXPERIMENTATION RESULT AND EVALUATION

### 5.1 COMPARISON OF LOB MID-PRICE PREDICTION

#### 5.1.1 EXPERIMENT SETTING FOR LOB MID-PRICE PREDICTION

**Dataset** All the experiments are based on cryptocurrency LOB data from Binance (https://www.binance.com) websocket API. In this experiment, one-day LOB data of product BTC-USDT (Bitcoin-U.S. dollar tether) on 2022.07.15. containing 863397 ticks. The time interval between each ticks is not evenly spaced. The time interval is $0.1$ second on average. The first $70\%$ data is used to construct the training set, and the rest $10\%$ and $20\%$ of data are used for validation and testing.

**Models** For the comparison purpose, canonical LSTM and vanilla Transformers along with four Transformer-based models are choosed: FEDformer Zhou et al. (2022), Autoformer Wu et al. (2021), Informer Zhou et al. (2020) and Reformer Kitaev et al. (2020).

**Training setting** The dataset is normalized by the z-score normalization method. All the models are trained for 10 epochs using the Adaptive Momentum Estimation optimizer and L2 loss with early stopping. The batch size is 32, and the initial learning rate is 1e-4. All models are implemented by Pytorch Paszke et al. (2019) and trained on a single NVIDIA RTX A5000 GPU with 24 GB memory with AMD EPYC 7551P CPU provided from gpushare.com cluster.

#### 5.1.2 RESULT AND ANALYSIS FOR LOB MID-PRICE PREDICTION

**Quantitative result** The performance metrics consist of Mean Square Error (MSE) and Mean Absolute Error (MAE). From the table 1, these outcomes can be summarized: In a comparison of different models, both FEDformer and Autoformer demonstrate superior performance over LSTM, with FEDformer achieving the best results across all prediction lengths. Specifically, FEDformer reduces mean squared error (MSE) by $24\%$ from $0.104$ to $0.0793$ for a 96 prediction length and $21\%$

| Models | FEDformer | | Autoformer | | Informer | | Reformer | | Transformer | | LSTM | |
|---|---|---|---|---|---|---|---|---|---|---|---|---|
| Metrics | MSE | MAE | MSE | MAE | MSE | MAE | MSE | MAE | MSE | MAE | MSE | MAE |
| 96 | **0.0793** | **0.179** | 0.0926 | 0.201 | 1.411 | 0.543 | 2.186 | 0.619 | 2.836 | 0.696 | 0.104 | 0.204 |
| 192 | **0.155** | **0.257** | 0.176 | 0.279 | 1.782 | 0.749 | 1.842 | 0.824 | 2.799 | 0.832 | 0.195 | 0.287 |
| 336 | **0.274** | **0.348** | 0.319 | 0.376 | 2.080 | 0.830 | 9.218 | 1.947 | 1.456 | 0.665 | 0.315 | 0.369 |
| 720 | **0.608** | **0.514** | 0.643 | 0.539 | 2.808 | 1.093 | 72.57 | 6.824 | 4.306 | 1.297 | 0.771 | 0.587 |

Table 1: Mid price prediction result with different prediction lengths $k \in \{96, 192, 336, 720\}$ in test set. The input window size is set to 96 (MSE's unit is in $10^{-2}$ and MAE's unit is in $10^{-1}$; lower is better)

from 0.771 to 0.608 for a 336 prediction length, while Autoformer shows an 11% and 16% reduction in MSE for the same prediction lengths, respectively. This indicates their robustness and efficiency in reducing errors over long-term forecasts. Although LSTM does not perform as well as FEDformer and Autoformer, it still surpasses Informer, Reformer, and the vanilla Transformer in mid-price prediction tasks, suggesting that LSTM retains its robustness where transformer-based models falter without significant modifications. The vanilla Transformer and Reformer models exhibit poorer performance at various prediction lengths, attributed to error accumulation in the iterative multi-step (IMS) prediction process, and Informer's subpar performance is primarily due to its sparse attention mechanism, which leads to significant information loss in the time series.

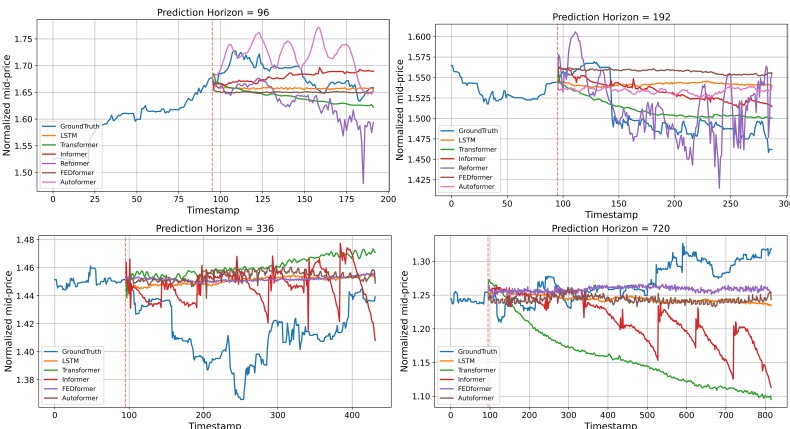

Figure 2: Illustration of normalized forecasting outputs with 96 input window size and $\{96, 192, 336, 720\}$ prediction lengths. Each timestamp is one tick.

**Qualitative Results and Limitations** Despite Autoformer and FEDformer demonstrating superior MSE and MAE performance compared to LSTM, their practical efficacy for high-frequency trading is questionable. Figure 2 illustrates the prediction results of various models across multiple horizons. While Autoformer and Reformer can model future mid-price trends at a 96 horizon, most models generate nearly flat predictions. At a 192 horizon, predictions generally plateau, with Reformer's outputs becoming more stochastic, and at longer horizons of 336 and 720, no model successfully predicts trends. This is further evidenced by the negative out-of-sample $R^2$ values for all models, as shown in Table 2, indicating that none of the models effectively explain the variance in mid-price based on the inputs used. The negative $R^2$ values highlight that the models are not adding value to the predictions. This discrepancy underscores the limitation of relying solely on MSE and MAE for evaluating model performance. Even models with favorable error metrics may fail to provide actionable predictions for trading, suggesting a potential shift towards using direct price difference as the target for more accurate and practical forecasting, which reveals that, while MSE and MAE metrics may indicate lower error, they can disguise the true limitations of models in Mid-Price Prediction.

| Models | Autoformer | FEDformer | Informer | Reformer | LSTM | Transformer |
|--------|-----------|-----------|----------|----------|------|-------------|
| 96 | -0.753 | -0.237 | -43.811 | -69.080 | -0.946 | -87.899 |
| 192 | -0.596 | -0.205 | -25.281 | -26.792 | -0.644 | -43.368 |
| 336 | -1.032 | -0.364 | -20.123 | -63.252 | -0.414 | -13.035 |
| 720 | -0.521 | -0.189 | -7.760 | -137.322 | -0.589 | -16.314 |

Table 2: Average of out of sample $R^2$ result with different prediction lengths $k \in \{96, 192, 336, 720\}$.

## 5.2 COMPARISON OF LOB MID-PRICE DIFF PREDICTION

### 5.2.1 EXPERIMENT SETTING FOR LOB MID-PRICE DIFF PREDICTION

**Dataset** The dataset for this experiment, has been expanded to four days of LOB data for BTC-USDT from July 3 to July 6, 2022, totaling 3,432,211 ticks, to mitigate overfitting. The first $80\%$ of data is used as a training set, and the rest $20\%$ is split in half for validation and testing.

**Models** Five models are being compared in this experiment: canonical LSTM Hochreiter and Schmidhuber (1997), vanilla transformer Vaswani et al. (2017), CNN-LSTM (DeepLOB Zhang et al. (2019) model used for regression), Informer Zhou et al. (2020) and Reformer Kitaev et al. (2020).

**Training settings** The training setting is the same as the last experiment.

### 5.2.2 RESULT AND ANALYSIS FOR LOB MID-PRICE DIFF PREDICTION

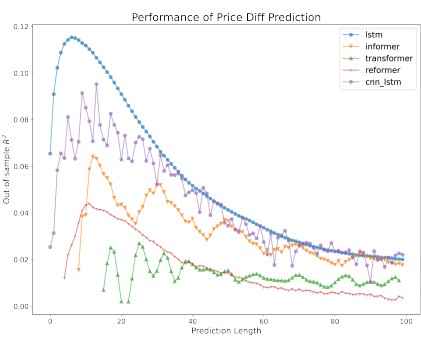

Figure 3: Performance of price difference prediction with input window size 100 and prediction length 100. Negative data points are not plotted for ease of visualization.

Following the previous works Kolm et al. (2021), out of sample $R^2$ is the evaluation metric for this task. The performance of all the models is shown in Figure 3. The canonical LSTM achieves the best performance among all models, which reaches the highest $R^2$ around $11.5\%$ in forecast length $5$ to $15$. For CNN-LSTM, it has comparable performance to LSTM. On the other hand, Informer, Reformer and Transformer have worse $R^2$ than LSTM, but their $R^2$ trend is similar. In short, for the price difference prediction task, LSTM-based models is more stable and more robust than Transformer-based models. In order to let these state-of-the-art transformer-based models make a meaningful prediction, a new structure is designed in the next part, and it is applied to the price movement prediction task.

## 5.3 COMPARISON OF LOB MID-PRICE MOVEMENT PREDICTION

### 5.3.1 INNOVATIVE ARCHITECTURE ON TRANSFORMER-BASED METHODS

For the task of predicting mid-price movements, where models classify future outcomes, few existing Transformer models are specifically designed, as most are oriented towards non-forecasting classification tasks. To bridge this gap, Transformer-based models have been adapted to enhance their capability in price movement forecasting by incorporating both past and projected mid-price data. This adaptation involves feeding a sequence of predicted mid-prices into a linear layer, followed by a softmax activation function to determine price movements. This approach, illustrated in Figure

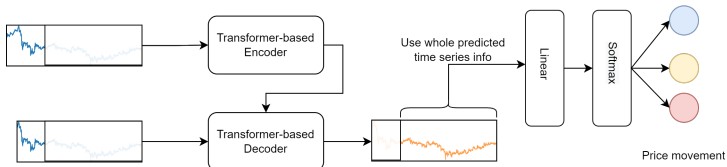

Figure 4: New architecture of transformer-based model for LOB mid-price movement prediction.

4, proves particularly effective with models using the Direct Multi-step (DMS) forecasting method, as it reduces long-term prediction errors and improves overall forecasting accuracy. This strategic enhancement is aimed at refining Transformer applications in financial forecasting.

### 5.3.2 DLSTM: INNOVATION ON LSTM-BASED METHODS

Inspired by the Dlinear model Zeng et al. (2022) and Autoformer, the DLSTM model combines time series decomposition with LSTM to leverage the strengths of both approaches. DLSTM capitalizes on three key observations: the effectiveness of time decomposition in enhancing forecasting performance as demonstrated in prior works Zhang et al. (2019); Wu et al. (2021); Zhou et al. (2022), the robustness of LSTM in handling diverse forecasting tasks, and Dlinear's success over other Transformer-based models in long time series forecasting due to its decomposition and DMS prediction methods. The architecture of DLSTM, which replaces the linear layers with LSTM layers as shown in Figure 5, incorporates a dual-layer approach where the time series $X_T = (x_1, x_2, \ldots, x_T)$ is first decomposed into a Trend series using a moving average:

$$X_t = AvgPool(Padding(X_T)) \tag{12}$$

where $AvgPool(\cdot)$ is the average pooling operation and $Padding(\cdot)$ is used to fix the input length. The Remainder series is calculated by $X_r = X_T - X_t$. After that, these two series are processed by separate LSTM layers, whose outputs are combined and passed through a linear and softmax activation to predict price movements, effectively handling one-step-ahead predictions without the error accumulation typically seen in multi-step forecasting.

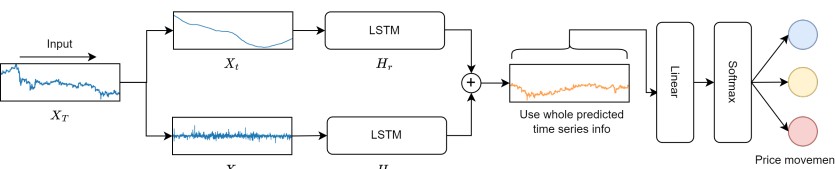

Figure 5: Architecture of DLSTM

### 5.3.3 SETTING FOR LOB MID-PRICE MOVEMENT PREDICTION

**Dataset** In this experiment, a dataset comprising 12 days of LOB data for ETH-USDT from July 3 to July 14, 2022, with 10,255,144 ticks. The training and testing data are taken from the first six days and the last three days, and the left data are used for validation. The test set is also used for the simple trading simulation.

**Models** Most of the transformer-based models are adapted in this task according to innovative structure in Section 5.3.1, which are: Vanilla Transformer Vaswani et al. (2017), Reformer Kitaev et al. (2020), Informer Zhou et al. (2020), Autoformer Wu et al. (2021), FEDformer Zhou et al. (2022). On the other hand, all the LSTM-based models are compared in this task as well, which are: canonical LSTM Hochreiter and Schmidhuber (1997), DLSTM, DeepLOB Zhang et al. (2019), DeepLOB-Seq2Seq Zhang and Zohren (2021), DeepLOB-Attention Zhang and Zohren (2021).

**Training settings** The batch size for training is set to 64 and the loss function is changed to Crossentropy loss. Other training settings are the same as the last experiment.

| Model | Acc | Prec | Rec | F1 | Acc | Prec | Rec | F1 |
|---|---|---|---|---|---|---|---|---|
| | **Prediction Horizon k = 20** | | | | **Prediction Horizon k = 30** | | | |
| MLP | 61.58 | 61.70 | 61.58 | 61.47 | 59.19 | 59.30 | 58.70 | 58.48 |
| LSTM | 62.77 | 62.91 | 62.77 | 62.78 | 60.64 | 60.47 | 60.45 | 60.45 |
| DeepLOB | 70.29 | 70.58 | 70.30 | 70.24 | 67.23 | 67.26 | 67.17 | 67.15 |
| DeepLOB-Seq2Seq | 70.40 | 70.79 | 70.42 | 70.37 | 67.56 | 67.73 | 67.53 | 67.49 |
| DeepLOB-Attention | 70.04 | 70.26 | 70.03 | 70.01 | 67.21 | 67.39 | 66.98 | 66.96 |
| Autoformer | 68.89 | 68.99 | 68.89 | 68.91 | 67.93 | 67.86 | 67.77 | 67.77 |
| FEDformer | 65.37 | 65.70 | 65.37 | 65.20 | 66.57 | 66.44 | 66.05 | 65.83 |
| Informer | 68.71 | 68.82 | 68.72 | 68.71 | 65.41 | 65.33 | 65.14 | 65.13 |
| Reformer | 68.01 | 68.26 | 68.00 | 67.95 | 64.28 | 64.31 | 64.08 | 64.06 |
| Transformer | 67.80 | 67.99 | 67.81 | 67.77 | 64.25 | 64.16 | 64.13 | 64.13 |
| DLSTM | **73.10** | **74.01** | **73.11** | **73.11** | **70.61** | **70.83** | **70.63** | **70.59** |
| | **Prediction Horizon k = 50** | | | | **Prediction Horizon k = 100** | | | |
| MLP | 55.65 | 55.71 | 55.62 | 54.98 | 57.03 | 56.03 | 56.36 | 56.01 |
| LSTM | 58.26 | 57.52 | 57.54 | 57.03 | 53.49 | 52.83 | 52.82 | 52.36 |
| DeepLOB | 63.32 | 63.69 | 63.32 | 63.37 | 58.12 | 58.50 | 57.92 | 57.86 |
| DeepLOB-Seq2Seq | 63.62 | 64.04 | 63.61 | 63.59 | 58.30 | 58.43 | 57.93 | 57.77 |
| DeepLOB-Attention | 64.05 | 64.19 | 64.04 | 63.94 | 59.16 | 58.59 | 58.65 | 58.50 |
| Autoformer | 60.17 | 60.64 | 60.12 | 58.40 | 59.18 | 58.34 | 58.40 | 57.83 |
| FEDformer | 63.46 | 63.44 | 63.42 | 62.52 | 57.97 | 56.97 | 56.62 | 54.14 |
| Informer | 61.76 | 61.64 | 61.74 | 61.55 | 56.11 | 56.15 | 55.85 | 55.81 |
| Reformer | 60.43 | 60.79 | 60.42 | 60.37 | 54.92 | 54.47 | 54.53 | 54.47 |
| Transformer | 59.51 | 59.78 | 59.51 | 59.46 | 55.42 | 55.04 | 54.92 | 54.72 |
| DLSTM | **67.45** | **67.96** | **67.45** | **67.59** | **63.73** | **63.02** | **63.18** | **63.05** |

Table 3: Experiment results of Mid Price Movement for prediction horizons 20, 30, 50 and 100. **Red Bold** represents the best result and blue underline represents the second best result.

### 5.3.4 RESULT AND ANALYSIS FOR LOB MID-PRICE MOVEMENT PREDICTION

The models' performance, evaluated using classification metrics including accuracy, precision, recall, and F1-score, is displayed in Tables 3. DLSTM surpasses all previous LSTM-based and Transformer-based models across all prediction horizons, demonstrating the effectiveness of integrating Autoformer's time series decomposition structure with a simple LSTM model for one-step-ahead predictions, thereby avoiding error accumulation typical in DMS processes. The DeepLOB-Attention model performs well at the 50 and 100 horizons, and the DeepLOB-Seq2Seq excels at the 20 horizon, highlighting the benefits of encode-decoder structures and attention mechanisms in capturing correlations across different prediction horizons. While the performance of DeepLOB-Attention and DeepLOB-Seq2Seq either matches or exceeds DeepLOB, particularly over longer horizons, Autoformer ranks second at the 30 horizon, underscoring its utility in time series prediction despite its size and tuning requirements compared to the more compact and less parameter-sensitive LSTM models.

### 5.3.5 SIMPLE TRADING SIMULATION WITHOUT TRANSACTION COST

To demonstrate the practical utility of the models in trading, a simple trading simulation (backtesting) is conducted using three high-performing models: DLSTM, DeepLOB Zhang et al. (2019), and Autoformer Wu et al. (2021), with Canonical LSTM Hochreiter and Schmidhuber (1997) and Vanilla Transformer Vaswani et al. (2017) serving as baselines. The simulation, conducted over a three-day test set, follows strategy from prior research Zhang et al. (2019). It involves trading a single share ($\mu = 1$) based on the model's prediction of price movements (0 for fall, 1 for stationary, 2 for rise). A long position is initiated at 'rise' and held until a 'fall' prediction occurs; conversely, a short position starts at 'fall'. To mimic high-frequency trading latency, a five-tick delay is implemented between prediction and execution. Only one position direction is allowed at any time in the simulation.

Table 4 show the profitability of each model in simulated trading, evaluated by cumulative price return (CPR) and the Annualized Sharpe Ratio (SR). The exaggerated value of the annualized SR results from the overly optimistic assumptions of the simulation. Results indicate that LSTM-based models generally outperform Transformer-based models in trading simulations. The canonical LSTM model

| Forecast Horizon | Prediction Horizon = 20 | | Prediction Horizon = 30 | | Prediction Horizon =50 | | Prediction Horizon=100 | |
|---|---|---|---|---|---|---|---|---|
| Model | CPR | SR | CPR | SR | CPR | SR | CPR | SR |
| LSTM | **15.396** | 51.489 | 12.458 | 41.411 | **8.484** | **28.817** | **4.914** | **20.941** |
| DLSTM | 14.966 | 46.949 | 12.634 | 37.432 | 6.194 | 22.027 | 3.215 | 16.346 |
| DeepLOB | 13.859 | 56.094 | **12.789** | **42.567** | 5.726 | 21.014 | 2.646 | 14.992 |
| Transformer | 14.553 | **59.995** | 12.737 | 41.044 | 6.896 | 28.147 | 2.859 | 16.981 |
| Autoformer | 9.942 | 32.688 | 8.617 | 30.576 | 8.214 | 25.882 | 3.620 | 17.765 |

Table 4: Cumulative price returns and annualized sharpe ratio of different models.

records the highest CPR and SR at the 20 and 30 horizons, while DeepLOB excels at the 50 horizon. DLSTM shows performance comparable to both canonical LSTM and DeepLOB. Autoformer, despite its superior classification metrics, underperforms in the 20 and 30 horizons, even lagging behind the vanilla Transformer, underscoring the relative effectiveness of LSTM-based models for electronic trading.

DLSTM demonstrates performance commensurate with these models, underscoring the practicality and robustness of LSTM-based predictions for trading. Conversely, Autoformer underperforms at the 20 and 30 horizons, sometimes even lagging behind the vanilla Transformer despite better classification metrics, highlighting LSTM-based models as more effective for electronic trading.

### 5.3.6 SIMPLE TRADING SIMULATION WITH TRANSACTION COST

| Forecast Horizon | Prediction Horizon = 20 | | Prediction Horizon = 30 | | Prediction Horizon =50 | | Prediction Horizon=100 | |
|---|---|---|---|---|---|---|---|---|
| Model | CPR | SR | CPR | SR | CPR | SR | CPR | SR |
| LSTM | 2.102 | 15.160 | 1.767 | 12.429 | 1.596 | 11.536 | 0.778 | 6.014 |
| DLSTM | **3.039** | **19.962** | **2.716** | **16.523** | **1.957** | **12.359** | **1.180** | **9.811** |
| DeepLOB | 1.964 | 15.082 | 1.924 | 13.128 | 1.450 | 10.273 | 0.823 | 7.993 |
| Transformer | 1.860 | 13.894 | 1.561 | 10.917 | 1.047 | 6.612 | 0.118 | -23.496 |
| Autoformer | 0.189 | -8.704 | 0.873 | 5.118 | -0.225 | -9.193 | -0.061 | -14.835 |

Table 5: Cumulative price returns and annualized sharpe ratio of different models under $0.002\%$ transaction cost.

Introducing a hypothetical transaction cost of $0.002\%$ in the simulation reveals that DLSTM consistently outperforms all models across all prediction horizons, demonstrating its profitability and robustness even with transaction costs factored in, as shown in Table 5. While LSTM-based models generally outperform Transformer-based ones, with Canonical LSTM and DeepLOB achieving competitive CPRs and SRs, Transformer models, particularly Autoformer, suffer significant performance drops, yielding negative returns in some cases.

## 6 CONCLUSION

This study conducts a comprehensive comparison of LSTM-based and Transformer-based models on three cryptocurrency LOB data prediction tasks. In the first task of predicting the LOB mid-price, FEDformer and Autoformer demonstrate lower error rates than other models, although LSTM outperforms Informer, Reformer, and vanilla Transformer. Despite lower prediction errors, the practical utility of these results for high-frequency trading is limited due to insufficient quality. In the second task of predicting the mid-price difference, LSTM-based models showcase superior robustness and performance, achieving the highest $R^2$ of 11.5% within about 10 prediction steps, while state-of-the-art models like Autoformer and FEDformer falter due to their inability to effectively process difference sequences.

For the final task, predicting LOB mid-price movement, a novel DLSTM model integrating LSTM with Autoformer's time decomposition architecture significantly outshines all models in classification metrics, proving its efficacy in trading simulations, especially under transaction costs. Overall, while Transformer-based models may excel in limited aspects of mid-price prediction, LSTM-based models demonstrate consistent superiority across the board, reaffirming their robustness and practicality in financial time series prediction for electronic trading.

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
