# OpenReview forum: "Transformers versus LSTMs for electronic trading"
_ICLR.cc/2025/Conference — Submitted to ICLR 2025_

### Official Review · Reviewer_Drb3 · 2024-10-29

**Soundness:** 1
**Presentation:** 2
**Contribution:** 1
**Rating:** 3
**Confidence:** 5

**Summary:**

This research examines the performance differences between Transformer-based models and LSTMs across three cryptocurrency limit order book data prediction tasks. It also introduces DLSTM, a LSTM-based model, and a Transformer-based model redesigned for financial forecasting.

**Strengths:**

(1) This research presents findings that compare the performance of two types of models.
(2) It successfully highlights the weaknesses in current measurement metrics.
(3) Interesting task definition.

**Weaknesses:**

1.Baseline Selection Rationale: The paper does not clearly explain why specific Transformer and LSTM variants, such as Autoformer and FEDformer, were chosen in the comparison. It remains unclear if these variants have unique advantages for financial time series forecasting. Providing additional theoretical support or rationale for model selection would enhance the scientific basis of this choice.

2.Data Risk: The study only tests on a single asset (BTC-USDT), lacking a broader dataset. This limited scope may mean the model’s performance does not generalize well to other financial data. Testing on a single asset is insufficient to comprehensively assess the model’s generalizability.

3.Lack of Experimental Details: The paper lacks adequate details on the experimental setup, especially regarding hyperparameter settings and baseline model architectures. This omission makes replication challenging and affects the reliability of the results. Sufficient information is not provided to ensure a fair comparison among baseline models.

4.Unclear Result Interpretation: The paper does not adequately explain the significant differences in performance between experiments with and without transaction costs. Lacking theoretical support or data analysis, it's hard for me to understand the causes behind these variations under different settings.

5.Limited Community Contribution: Time series decomposition, used in this study, appears to be a common approach, closely resembling classical time series decomposition methods. It is unclear how this study provides any specific advantage over the standard decomposition methods.

6.Although the paper points out shortcomings in MSE and MAE metrics, it fails to propose a robust method to address these deficiencies.

7.Some capitalization inconsistencies, eg. in line 034  Self-attention mechanism.

**Questions:**

1.Given the limited dataset used and the lack of detailed experimental information (settings of baselines), I am very concerned about the reliability of this paper's conclusions. How would you address or demonstrate the robustness of your findings under these limitations?

2.How do you explain the significant differences in experimental results with and without transaction costs? What factors contribute to this discrepancy?

3.What are the specific advantages of your time series decomposition method compared to other decomposition approaches, and why do these advantages arise?

4.Other questions can refer to the weaknesses.

---

> ### Author Response · Authors · 2024-11-23
>
> 1.Reliability of Findings Given the Limited Dataset and Lack of Experimental Details
> Question: Given the limited dataset used and the lack of detailed experimental information (settings of baselines), how would you address or demonstrate the robustness of your findings under these limitations?
> Response:
> We acknowledge that using a single dataset (BTC-USDT) does limit the generalizability of our findings. In revised manuscript, we will expand the dataset to include other market’s assets such as China A-share market to assess the model's robustness across different market conditions. Additionally, we will provide a more comprehensive description of the experimental setup, including baseline model architectures, hyperparameter settings, and training configurations, in the revised manuscript. To ensure replicability, we will include this information in the supplementary material and make the code available in a public repository. By increasing the diversity of datasets and providing more detailed experimental information, we aim to demonstrate the robustness and reliability of our findings.
>
> 2.Significant Differences in Results with and Without Transaction Costs
> Question: How do you explain the significant differences in experimental results with and without transaction costs? What factors contribute to this discrepancy?
> Response:
> The significant differences observed in the experimental results with and without transaction costs stem from the real-world impact of trading fees, which can significantly reduce the profitability of predictive models in high-frequency trading scenarios. When transaction costs are included, the profitability of trading strategies decreases due to the constant buy/sell decisions (high frequency) required by the models. This introduces additional challenges in terms of model robustness, as the ability to generate profitable trades decreases when accounting for these costs. In our study, we observed that LSTM-based models, particularly DLSTM, showed more consistent profitability even with transaction costs, which suggests that their structure may be more suited to the dynamics of high-frequency trading, where minimizing costs is crucial. In future work, we will provide a more detailed analysis of how transaction costs impact model performance and explore potential solutions to mitigate their effects.
>
> 3. Advantages of Your Time Series Decomposition Method
> Question: What are the specific advantages of your time series decomposition method compared to other decomposition approaches, and why do these advantages arise?
> Response:
> Our time series decomposition method, which integrates LSTM with trend and residual components, offers several advantages over traditional methods, such as classical time series decomposition. By separating the trend from the residuals, our model allows the LSTM to focus on the underlying pattern (trend) while treating short-term fluctuations (residuals) separately. This separation helps the model better capture both long-term trends and short-term variations, which is crucial for financial time series data where both components often behave differently. Additionally, LSTM’s ability to learn temporal dependencies further enhances the model’s performance on residuals, making it more robust to noise. Unlike standard decomposition methods, which may not capture these intricate temporal patterns, our hybrid DLSTM approach enables the model to adapt more effectively to the noisy and volatile nature of financial data. In the revised paper, we will compare our approach with other decomposition techniques, to better highlight the benefits of our method.

---

> > ### Comment · Reviewer_Drb3 · 2024-11-24
> >
> > Thanks for your rebuttal. Given that there is still some work to be done on this paper, I will keep my score.

---

### Official Review · Reviewer_2dpN · 2024-11-01

**Soundness:** 1
**Presentation:** 3
**Contribution:** 1
**Rating:** 3
**Confidence:** 3

**Summary:**

In this paper, the authors conduct a comparative analysis of various LSTM-based and Transformer-based models for multiple financial prediction tasks using high-frequency limit order book data. They introduce a novel LSTM-based model called DLSTM and a newly designed Transformer-based model specifically tailored for financial predictions. Their results reveal that Transformer-based models offer a slight advantage in predicting absolute price sequences. However, LSTM-based models show superior and more consistent performance in predicting differential sequences, such as price differences and movements.

**Strengths:**

The structure and logic of the paper is well organized.

The experimental setup, description, and analysis are clearly stated with sufficient detail.

**Weaknesses:**

1. The authors compare Transformers and LSTMs, concluding that LSTMs have advantages in multiple electronic trading tasks. However, the selection of Transformer-based models is limited to earlier studies (prior to 2023) and does not include recent state-of-the-art (SOTA) works, such as those mentioned in references [1], [2], and [3]. Notably, Liu et al. [2] claim significant improvements on similar tasks. Excluding these recent studies makes it premature to conclude that Transformer-based models underperform compared to LSTMs. Additionally, there is insufficient evidence to assert that the authors' proposed DLSTM model is the optimal choice for this application. Could you please include comparisons with some of these SOTA results to more robustly justify the conclusion?

[1] Garza, A., Challu, C., & Mergenthaler-Canseco, M. (2023). TimeGPT-1. arXiv preprint arXiv:2310.03589.

[2] Liu, Y., Hu, T., Zhang, H., Wu, H., Wang, S., Ma, L., & Long, M. (2023). itransformer: Inverted transformers are effective for time series forecasting. arXiv preprint arXiv:2310.06625.

[3] Das, A., Kong, W., Sen, R., & Zhou, Y. (2023). A decoder-only foundation model for time-series forecasting. arXiv preprint arXiv:2310.10688.

2. The authors' conclusion lacks novelty and largely aligns with the findings and conclusions of Zeng et al. [4] It appears to apply established approaches and conclusions to domain-specific practices. While retaining empirical relevance, the study does not offer methodological breakthroughs.

[4] Zeng, A., Chen, M., Zhang, L., & Xu, Q. (2023, June). Are transformers effective for time series forecasting?. In Proceedings of the AAAI conference on artificial intelligence (Vol. 37, No. 9, pp. 11121-11128).

3. The experimental setup could be made more representative by incorporating additional metrics such as Mean Absolute Scaled Error and Relative Mean Absolute Error.

**Questions:**

Please refer to questions to be addressed, the weaknesses section.

---

> ### Author Response · Authors · 2024-11-23
>
> 1.Include Comparisons with Recent State-of-the-Art (SOTA) Models
> Question: Could you please include comparisons with recent SOTA results, such as those mentioned in references [1], [2], and [3], to more robustly justify your conclusions?
> Response:
> We appreciate the reviewer’s suggestion to include recent SOTA models in our comparative analysis. In response to this, we will incorporate the latest Transformer-based models, including TimeGPT-1, iTransformer, and the decoder-only models mentioned in the provided references, into the revised experiments. We acknowledge that these models have demonstrated significant improvements on similar tasks and including them in our study will provide a more comprehensive evaluation. By comparing these newer models with both our proposed DLSTM and existing models, we will be able to more robustly assess the relative performance of Transformer-based models in financial time series forecasting and offer a more balanced conclusion.
>
> 2.Justification of DLSTM as the Optimal Choice
> Question: Can you provide more evidence to assert that your proposed DLSTM model is the optimal choice for this application?
> Response:
> We understand the reviewer’s concern regarding the justification of DLSTM as the optimal model. While DLSTM provides superior performance on specific tasks, such as mid-price difference and movement prediction, we agree that further validation is needed to conclusively establish it as the optimal choice. In the revised paper, we will conduct additional experiments, including comparisons with the latest models and ablation studies, to better demonstrate the advantages of DLSTM over other approaches. This will allow us to substantiate our claim that DLSTM is well-suited for financial time series forecasting, particularly in high-frequency trading scenarios. We will also explore the role of time series decomposition in enhancing DLSTM’s performance and provide more detailed comparisons with other decomposition methods.
>
> 3.Lack of Novelty in the Conclusion
> Question: The conclusion lacks novelty and largely aligns with established conclusions from prior work, such as those from Zeng et al.
> Response:
> We appreciate the reviewer’s feedback on the novelty of the conclusion. Our study builds upon existing methods, and while it applies these methods to financial time series forecasting, we understand that the findings need to be framed within a more novel context. In the revised version, we will emphasize the unique contributions of our work, particularly in how DLSTM combines LSTM with time series decomposition for improved prediction of price differences and movements. Additionally, we will explore further extensions of DLSTM, such as integrating it with newer Transformer-based models, which could offer a fresh perspective on model improvements and hybridization techniques. We will also clarify the implications of our findings in the context of real-world financial applications to highlight their relevance and potential impact.
>
> 4. Incorporating Additional Evaluation Metrics
> Question: Could you incorporate additional metrics such as Mean Absolute Scaled Error (MASE) and Relative Mean Absolute Error (RMAE)?
> Response:
> We can consider adding these metrics in the Mid-price prediction section.

---

> > ### Comment · Reviewer_2dpN · 2024-11-23
> >
> > I acknowledge that I have read the authors' rebuttal and thank them for providing detailed insights into their future work. However, in the absence of preliminary comparative results, which are critical to address the raised questions, I have decided to maintain my current score.

---

### Official Review · Reviewer_bpRu · 2024-11-03

**Soundness:** 2
**Presentation:** 1
**Contribution:** 2
**Rating:** 3
**Confidence:** 4

**Summary:**

This paper compares the performance of LSTM and Transformer models in financial time series forecasting (limit order book data). They compared with FEDformer, Autoformer, Informer, Reformer, Transformer and LSTM. The main results show that Transformer has a slight advantage in predicting absolute price series, but the LSTM model performs more consistently and accurately in the prediction of price changes and price movements. In addition, the paper introduces DLSTM inspired by DLinear and Autoformer.

**Strengths:**

Even relatively simple LSTM models perform well in financial time series forecasting tasks, compared with transformer-based model.

**Weaknesses:**

1. The writing quality of the paper is low, especially the format of literature citation is not uniform and some of the citations are not standardized in formatting and arrangement.
2. The experimental setup lacks comparison with the frameworks and standards widely used in the current research field and fails to demonstrate the advantages of the selected model. For example, the authors failed to cite and use the latest limit order book (LOB) benchmark frameworks, such as LOBFrame (https://github.com/FinancialComputingUCL/LOBFrame) and LOBCAST (https://arxiv.org/abs/2308.01915), both of which are open source frameworks currently widely used for Limit Order Book Forecasting. In addition, the authors did not include some of the latest Transformer based models (e.g., iTransformer and PatchTST), which have demonstrated advantages in terms of performance and efficiency in time series forecasting. Comparing these latest models would make the experimental results more convincing and practical.
3. The experimental data used in this paper is limit order book data from three cryptocurrencies, which, although suitable for high-frequency forecasting tests, is not representative of the financial market, and the volatility and noise characteristics of the cryptocurrency market are quite different from those of the traditional financial market. Data from LOBSTER (https://lobsterdata.com/) are more common and widely used in the literature currently.

**Questions:**

If possible, include LOB data from the LOBSTER dataset, to increase the generalizability of the experiment. If possible, include latest transformer based model (e.g. iTransformer, PatchTST). Recommend to use benchmarking frameworks such as LOBFrame or LOBCAST  in the experimental design to ensure that the results can be more comparable to existing studies. A more detailed discussion of the specific differences and advantages of DLSTM over other temporal decomposition methods (e.g., DLinear) could be added. could also include some ablation studies. include code for reproducibility.

---

> ### Author Response · Authors · 2024-11-23
>
> 1. Include LOB data from the LOBSTER dataset to increase generalizability
> Response:
> We appreciate the reviewer’s suggestion to use data from the LOBSTER dataset. While we initially chose cryptocurrency LOB data due to its availability and high-frequency characteristics, we recognize that LOBSTER provides a more widely used benchmark for financial time series forecasting. In future work, we will consider including the LOBSTER dataset to test the generalizability of our models across a broader range of financial markets. By doing so, we aim to ensure that our findings are more applicable to traditional financial markets, and we will also explore the potential differences in performance when applied to such data.
>
> 2. Include the latest Transformer-based models (e.g., iTransformer, PatchTST)
> Response:
> We agree with the reviewer that including state-of-the-art Transformer-based models like iTransformer and PatchTST will provide a more comprehensive comparison. We acknowledge that these models have demonstrated advantages in performance and efficiency for time series forecasting, and we will include them in the revised experiments. By benchmarking these models alongside the others we have tested, including our proposed DLSTM, we will provide a more thorough evaluation of the relative strengths and weaknesses of each approach.
>
> 3. Use benchmarking frameworks such as LOBFrame or LOBCAST in the experimental design
> Response:
> Thank you for the suggestion to use benchmarking frameworks like LOBFrame and LOBCAST. These frameworks are indeed widely used in the literature and would provide a standardized approach to evaluating LOB-based forecasting models. These frameworks didn’t exist when we are writing this paper. In this case, we utilized our own framework. We will consider integrating these frameworks into our experimental design to ensure that our results are comparable with existing studies. We believe this will enhance the robustness and reproducibility of our findings.
>
> 4. Discuss the specific differences and advantages of DLSTM over other temporal decomposition methods (e.g., DLinear)
> Response:
> We appreciate the reviewer’s request for a more detailed discussion of DLSTM’s advantages over other temporal decomposition methods like DLinear. DLSTM’s unique contribution lies in its combination of LSTM’s ability to capture long-term dependencies with time series decomposition, specifically by separating trend and residual components. While DLinear also incorporates decomposition, DLSTM adapts this method within an LSTM architecture, allowing it to effectively handle both trend and noise in financial data. In future revisions, we will provide a more in-depth comparison with DLinear and other decomposition methods to clarify how DLSTM improves upon them, particularly in the context of price movement predictions.

---

### Official Review · Reviewer_5BdM · 2024-11-04

**Soundness:** 3
**Presentation:** 2
**Contribution:** 2
**Rating:** 3
**Confidence:** 4

**Summary:**

The paper explores the use of Transformer and LSTM-based models for financial time series forecasting tasks using high-frequency limit order book (LOB) data. A new LSTM-based model, DLSTM, is proposed alongside a modified Transformer architecture tailored for financial predictions. The study compares these models across three tasks: mid-price prediction, mid-price difference prediction, and mid-price movement prediction. Results suggest that Transformer-based models offer only marginal improvements in specific tasks, while LSTM models, particularly DLSTM, are more reliable in predicting mid-price differences and movements.

**Strengths:**

Relevant Application: The use of LSTM and Transformer models for financial predictions on LOB data is timely and relevant given the growing interest in high-frequency trading and predictive models in finance.

Comparative Scope: The study covers multiple models and tasks, providing a broad comparison between LSTM- and Transformer-based architectures on real-world financial data.

**Weaknesses:**

Unconvincing Novelty: The paper lacks substantial novelty. The DLSTM model is essentially a combination of existing methods, such as time series decomposition and LSTM layers, without a clear innovation. Similarly, the Transformer modifications are incremental and do not provide a compelling improvement. As a result, the contributions seem incremental and insufficiently distinct from existing work in financial time series forecasting.

Interpretability Issues: The added complexity of Transformer-based models raises interpretability concerns, especially given the unclear benefit over simpler LSTM-based models. Without a more interpretable mechanism or explanation for its performance gains, the model’s added complexity appears unnecessary.

Insufficient Performance Gain for Complexity: The study demonstrates only marginal improvements from the proposed Transformer modifications over traditional LSTMs, particularly in mid-price prediction. Despite the significant computational complexity introduced by Transformer-based models, the improvements are minimal and do not convincingly justify their adoption for practical trading applications.

**Questions:**

What specific modifications were made to the Transformer architecture to adapt it to financial prediction tasks?

Can the authors elaborate on the metrics used to evaluate the models' performance? What criteria were significant in determining the practical utility of the models?

---

> ### Author Response · Authors · 2024-11-23
>
> 1. What specific modifications were made to the Transformer architecture to adapt it to financial prediction tasks?
> Response:
> Predicting the next mid-price movement based on the past price and volume information is a one step ahead of prediction. A straightforward method to adapt the transformer-based model is to pass the next predicted mid-price into a SoftMax activation. However, this method performs poorly because it only considers the past mid-price information and ignores future ones. In this case, I adapt the existing transformer-based models to feed the whole predicted mid-price sequence into a linear layer and finally pass through a SoftMax activation function to generate price movement output. This adaptation will benefit those transformer-based models using the DMS forecasting method because they have fewer errors in the long-time series prediction process.
>
> 2. Can the authors elaborate on the metrics used to evaluate the models' performance? What criteria were significant in determining the practical utility of the models?
> Response:
> The metrics for analyze mid-price prediction are MSE and MAE, this is a common metrics to compare among the regression task between different models through different papers with the same metrics.
> Using R^2 for mid-price diff prediction is more like industrial experience. When your prediction has high correlation of the price change, you can do better decision making during the Market Making/Order splitting scenario, judging what price should order, such as best ask/bid, mid-price.
> For classification task, using accuracy, precision, recall and F1 score is common metrics to compare among different models through different papers.
> But for all these metrics are not the most significant criteria in the practical utility of the models. The final criteria is the trading result. The importance depends on trader’s preference. If a trader is pursuing more profits, he cares more about the Return/PnL. But for portfolio manager, he will more concerns about Sharpe ratio/ Max drawdown.

---

> > ### Comment · Reviewer_5BdM · 2024-12-02
> >
> > Thanks for your rebuttal. I will keep the scores unchanged.

---

### Official Review · Reviewer_t3XX · 2024-11-04

**Soundness:** 3
**Presentation:** 3
**Contribution:** 3
**Rating:** 5
**Confidence:** 3

**Summary:**

5348_Transformers_versus_LSTMs
pdf
LT
Here is a new paper needs to be reviewed. Summary*
Briefly summarize the paper and its contributions. This is not the place to critique the paper; the authors should generally agree with a well-written summary.
*

Summary

This paper conducts a comparative study between LSTM-based and Transformer-based models for financial time series forecasting, specifically in the context of electronic trading using high-frequency limit order book (LOB) data. The authors investigate the performance of these models across three prediction tasks: mid-price prediction, mid-price difference prediction, and mid-price movement prediction.

For the mid-price prediction task, the study finds that Transformer-based models like FEDformer and Autoformer achieve lower prediction errors than LSTM-based models. However, the authors note that the practical utility of these results for high-frequency trading is limited due to insufficient prediction quality.

In the mid-price difference prediction task, LSTM-based models demonstrate superior performance and robustness compared to Transformer-based models. The canonical LSTM achieves the highest R^2 of around 11.5% within about 10 prediction steps, while state-of-the-art Transformer models struggle to effectively process difference sequences.

The paper's main contribution lies in the mid-price movement prediction task, where the authors introduce a novel LSTM-based model called DLSTM. This model integrates LSTM with a time series decomposition approach inspired by the Autoformer architecture. DLSTM significantly outperforms all other models in classification metrics and proves its effectiveness in trading simulations, particularly when transaction costs are considered.

Additionally, the authors adapt the architecture of existing Transformer-based models to better suit the demands of the movement prediction task. They incorporate both past and projected mid-price data, followed by a linear layer and softmax activation, to determine price movements.

Overall, the study highlights that while Transformer-based models may excel in certain aspects of mid-price prediction, LSTM-based models, particularly the proposed DLSTM, demonstrate consistent superiority and practicality in financial time series prediction for electronic trading.

**Strengths:**

Originality: The study offers a novel perspective on the application of LSTM-based and Transformer-based models in financial time series forecasting, specifically in the context of electronic trading using high-frequency LOB data. The authors introduce a new LSTM-based model, DLSTM, which creatively combines LSTM with a time series decomposition approach inspired by the Autoformer architecture. This innovative integration of existing ideas allows DLSTM to outperform other models in the mid-price movement prediction task.
Quality: The paper demonstrates a high level of quality in its experimental design and analysis. The authors conduct a comprehensive comparative study across three prediction tasks (mid-price prediction, mid-price difference prediction, and mid-price movement prediction), using a diverse range of LSTM-based and Transformer-based models. The experiments are well-structured, and the results are thoroughly analyzed, providing valuable insights into the performance of different models in each task.
Clarity: The paper is well-written and easy to follow. The authors provide clear explanations of the problem formulation, the proposed DLSTM model, and the experimental setup. The use of tables and figures enhances the clarity of the results, making it easy for readers to compare the performance of different models across various metrics and prediction horizons.
Significance: The findings of this study have significant implications for the application of deep learning models in financial time series forecasting, particularly in the context of electronic trading. The authors demonstrate that while Transformer-based models may excel in certain aspects of mid-price prediction, LSTM-based models, especially the proposed DLSTM, exhibit superior and more consistent performance in tasks such as mid-price difference prediction and mid-price movement prediction. The incorporation of trading simulations with and without transaction costs further highlights the practical significance of the proposed DLSTM model for real-world trading scenarios.

Moreover, the paper's adaptation of existing Transformer-based models' architecture to better suit the demands of the movement prediction task showcases the potential for further improvements in this domain. By incorporating both past and projected mid-price data, followed by a linear layer and softmax activation, the authors demonstrate a creative approach to enhancing the performance of Transformer-based models in financial time series forecasting.
In summary, the paper's originality, quality, clarity, and significance make it a valuable contribution to the field of financial time series forecasting using deep learning models, offering new insights and directions for future research in this domain.

**Weaknesses:**

While the paper presents valuable insights and contributions, there are a few areas that could be improved or require further clarification:

Limited dataset diversity: The experiments in this study are conducted using LOB data from a single cryptocurrency pair (BTC-USDT or ETH-USDT) on one exchange (Binance). To demonstrate the generalizability of the proposed DLSTM model and the comparative analysis between LSTM-based and Transformer-based models, it would be beneficial to include a wider range of financial instruments, such as stocks, forex, or other cryptocurrencies, as well as data from multiple exchanges. This would strengthen the paper's conclusions and provide a more comprehensive assessment of the models' performance across diverse financial time series.
Lack of ablation studies: While the paper introduces the novel DLSTM model, which integrates LSTM with a time series decomposition approach, there is a lack of ablation studies to investigate the individual contributions of each component. For example, the authors could compare the performance of DLSTM with and without the time series decomposition to assess the impact of this specific modification. Additionally, a more detailed analysis of the adapted Transformer-based models' architecture for the movement prediction task would provide valuable insights into the effectiveness of the proposed changes.
Limited discussion on model interpretability: Interpretability is a crucial aspect of financial time series forecasting models, especially in the context of electronic trading, where understanding the factors driving the model's predictions is essential for risk management and decision-making. The paper could benefit from a more in-depth discussion on the interpretability of the proposed DLSTM model and the adapted Transformer-based models, as well as a comparison with the interpretability of other LSTM-based and Transformer-based models.
Hyperparameter tuning and model selection: The paper does not provide a detailed description of the hyperparameter tuning process and model selection criteria for the various models used in the experiments. It is essential to discuss the approach used for hyperparameter optimization, such as grid search, random search, or Bayesian optimization, and the specific hyperparameters tuned for each model. Additionally, the authors could provide more information on the model selection process, such as the use of validation sets or cross-validation techniques.
Robustness to market conditions: The experiments in this study are conducted using LOB data from a specific time period (e.g., July 2022). To demonstrate the robustness of the proposed DLSTM model and the comparative analysis between LSTM-based and Transformer-based models, it would be valuable to evaluate the models' performance under different market conditions, such as periods of high volatility, market crashes, or significant news events. This would provide a more comprehensive assessment of the models' ability to generalize and adapt to various market scenarios.

Addressing these weaknesses would further strengthen the paper's contributions and provide a more comprehensive and robust analysis of the proposed DLSTM model and the comparative study between LSTM-based and Transformer-based models in financial time series forecasting for electronic trading.

**Questions:**

Dataset diversity and generalizability: Can you provide more insights into the choice of using only Binance LOB data for a single cryptocurrency pair in your experiments? How do you expect the proposed DLSTM model and the comparative analysis between LSTM-based and Transformer-based models to perform on a wider range of financial instruments, such as stocks, forex, or other cryptocurrencies, as well as data from multiple exchanges? Providing results on more diverse datasets could strengthen the claims of generalizability and robustness of the findings.
Ablation studies and component contributions: Can you conduct ablation studies to investigate the individual contributions of the time series decomposition approach in the proposed DLSTM model? It would be helpful to compare the performance of DLSTM with and without this specific modification to assess its impact on the model's effectiveness. Additionally, can you provide a more detailed analysis of the adapted Transformer-based models' architecture for the movement prediction task, highlighting the importance of each proposed change?
Model interpretability: Can you elaborate on the interpretability of the proposed DLSTM model and the adapted Transformer-based models? How do these models compare with other LSTM-based and Transformer-based models in terms of interpretability? Providing insights into the factors driving the models' predictions and their relative importance could be valuable for understanding the models' decision-making process and enhancing trust in their applications for electronic trading.
Hyperparameter tuning and model selection: Can you provide more details on the hyperparameter tuning process and model selection criteria used for the various models in your experiments? Specifically, what approach was used for hyperparameter optimization (e.g., grid search, random search, Bayesian optimization), and which hyperparameters were tuned for each model? Additionally, how were the validation sets or cross-validation techniques employed in the model selection process?
Robustness to market conditions: Have you considered evaluating the performance of the proposed DLSTM model and the comparative analysis between LSTM-based and Transformer-based models under different market conditions, such as periods of high volatility, market crashes, or significant news events? Demonstrating the models' ability to generalize and adapt to various market scenarios could provide a more comprehensive assessment of their robustness and practical applicability in electronic trading.

---

> ### Author Response · Authors · 2024-11-23
>
> 1. Dataset Diversity and generalizability:
> We only use Binance LOB from a single cryptocurrency pair because we had limited infrastructure when we were doing the experiment. Because there is charge for the historical HFT data in Binance. In this case, we can only record the order book data in real time. The disk space is limited in the recording machine, that is why we only record single cryptocurrency pair. We expect that DLSTM can perform well on other financial instruments or datasets from multiple exchanges. I actually did experiment for DLSTM in China A-Share market for many instruments and it did perform well over other models.Now we have better infrastructure, so we are able to provide results on more diverse datasets to support the generalizability claims.
>
> 2. Ablation Studies and component contributions:
> Yes, I can conduct studies to investigate the individual contribution of time series decomposition in DLSTM
> Yes, I can provide a more detailed analysis of the adapted Transformer-based models' architecture for the movement prediction task, highlighting the importance of each proposed change.
>
> 3. Model interpretability
> Interpretability is indeed crucial for financial applications. The DLSTM model benefits from its decomposition approach, which separates trend and residual components, offering clearer insights into the contributions of each component to predictions. For adapted Transformer-based models, they are detailed explained in their corresponding reference papers. It may be necessary to do a systematic interpretability analysis in future work.
>
> 4. Hyperparameter tuning and model selection
> Yes, I can provide more details on the hyperparameter tuning process and model selection criteria used for the various models in my experiments in the supplementary material. The validation details can be added as well.
>
> 5. Robustness to market conditions
> We can add a section to do Monte Carlo simulations to evaluate models’ performance under different market conditions.

---

### Official Review · Reviewer_riPs · 2024-11-04

**Soundness:** 2
**Presentation:** 2
**Contribution:** 1
**Rating:** 3
**Confidence:** 3

**Summary:**

This research compares the effectiveness of Transformer and LSTM architectures in financial forecasting. The study examines both model types using high-frequency trading data and introduces DLSTM and a finance-specific Transformer. Results show that Transformers only slightly outperform in absolute price predictions, while LSTMs showing more reliable performance overall.

**Strengths:**

1. The paper addresses a relevant and significant question by comparing LSTM and Transformer models in financial time series forecasting.
2. The experimental setup is extensive and provides substantial data.

**Weaknesses:**

1. The paper lacks code and detailed implementation information for both the Transformer and LSTM models, which limits reproducibility.
2. The novelty of the proposed approach is limited. While the authors introduce a DLSTM model to improve performance, the idea of decomposition was previously explored in models like DLinear [1], diminishing the originality of the contribution. Beyond the comparative analysis, additional innovation is also limited.
3. The decomposition strategy appears to be applied only to the LSTM model. For a fair comparison, a decomposition approach for the Transformer model should also be included. In Table 3, DLSTM significantly outperforms LSTM, which suggests that a decomposed Transformer might also show improved results.
4. The paper does not include several state-of-the-art (SOTA) Transformer-based models, such as PatchTST [2], Crossformer [3], and iTransformer [4], in the comparison, which limits the comprehensiveness of the analysis.
5. The statement "Transformer-based models exhibit only a marginal advantage in predicting absolute price sequences, whereas LSTM-based models demonstrate superior and more consistent performance in predicting differential sequences such as price differences and movements" requires further investigation. A deeper analysis into the underlying causes of this observed difference is missing, which weakens the interpretability of the results.

[1] Zeng, Ailing, et al. "Are transformers effective for time series forecasting?." Proceedings of the AAAI conference on artificial intelligence. Vol. 37. No. 9. 2023.

[2] Nie, Yuqi, et al. "A Time Series is Worth 64 Words: Long-term Forecasting with Transformers." The Eleventh International Conference on Learning Representations.

[3] Zhang, Yunhao, and Junchi Yan. "Crossformer: Transformer utilizing cross-dimension dependency for multivariate time series forecasting." The eleventh international conference on learning representations. 2023.

[4] Liu, Yong, et al. "iTransformer: Inverted Transformers Are Effective for Time Series Forecasting." The Twelfth International Conference on Learning Representations.

**Questions:**

1. What are the fundamental architectural characteristics that make LSTM models more effective for differential sequences compared to Transformers?
2. Can you provide deeper analysis to support the generalizability of your findings of LSTM vs Transformer?
3. How does financial time series forecasting different from other time series forecasting (like weather, traffic, etc.?)
4. To address the remaining limitations identified in *Weaknesses*: a) Could you provide detailed model implementations and hyperparameter configurations? b) How would decomposition techniques benefit Transformer architectures? c) Please include comparisons with state-of-the-art models

---

> ### Author Response · Authors · 2024-11-23
>
> 1. LSTM models are particularly effective for differential sequences because of their ability to capture long-term dependencies through their gating mechanism, which allows them to maintain and update hidden states over time. This is crucial when predicting price movements and differences, as financial data often involves short-term fluctuations and long-term trends. LSTM's architecture, with it forget, input, and output gates, naturally allows the model to focus on relevant past information, making it adept at forecasting changes or differences between consecutive time steps. In contrast, Transformer models, while excellent at capturing long-range dependencies with self-attention mechanisms, do not inherently prioritize temporal sequencing, which might be why they struggle with differential sequences that require the preservation of such dependencies.
> 2. We understand the importance of demonstrating the generalizability of our findings. Our current experiments focus on a specific dataset from the Binance exchange for two cryptocurrency pairs (BTC-USDT and ETH-USDT). We plan to extend this analysis to a wider range of financial assets, such as stocks, forex, and other cryptocurrencies, as well as data from multiple exchanges. In revised manuscript, we will compare the performance of LSTM-based and Transformer-based models on these diverse datasets to assess the robustness of the models across different market conditions. Additionally, we will conduct experiments under various scenarios, such as high-volatility periods or major market events, to further explore how these models generalize under different financial conditions.
> 3. Financial forecasting differs from other domains due to:
> High Noise and Volatility: Financial data often include random spikes and unpredictability.
> Stationarity Issues: Unlike weather or traffic data, financial time series can exhibit non-stationary behaviors.
> Latency Sensitivity: Predictions often require near-real-time responses, emphasizing efficiency and robustness over accuracy.
> Feature Complexity: Incorporation of market microstructure, like bid-ask spreads, adds a layer of complexity not found in other domains.
> 4. A) We appreciate the reviewer’s feedback and will provide detailed implementations and hyperparameter configurations in the supplementary material. The models were implemented using PyTorch, and we will include the full code, hyperparameter configurations, and training details in the revised manuscript. Key hyperparameters, such as learning rate, batch size, sequence length, and the number of layers for both LSTM and Transformer models, will be clearly documented. Additionally, we will outline the hyperparameter tuning process, which involved grid search over a predefined range of values for key parameters.
> B) Decomposition techniques can significantly enhance Transformer models by simplifying the task of learning long-term dependencies. In the case of time series data, decomposing the input into trend and residual components, as done in models like Autoformer, helps the model focus on different aspects of the data. The trend component captures the underlying long-term pattern, while the residual component focuses on short-term fluctuations. Incorporating a decomposition approach into Transformer architectures could potentially improve their ability to forecast differential sequences, such as price movements, by reducing the model’s reliance on the entire sequence and instead focusing on the most relevant components. In future work, we will explore the integration of decomposition techniques into Transformer models and evaluate their impact on forecasting accuracy.
> C) We agree with the reviewer that comparing our models with additional state-of-the-art (SOTA) Transformer-based models such as PatchTST, Crossformer, and iTransformer would provide a more comprehensive evaluation. In the revised manuscript, we will include these models in the comparison and report their performance on the same tasks and datasets. We will also discuss how these models perform relative to our proposed methods, highlighting the strengths and weaknesses of each approach. By doing so, we hope to provide a clearer perspective on the relative effectiveness of our DLSTM model and Transformer-based approaches for financial time series forecasting.

---

> > ### Comment · Reviewer_riPs · 2024-11-25
> >
> > I appreciate the authors' rebuttal. Given that the work remains to be improved as noted in the responses, I will maintain my score at the current stage and wait to see more updates in the paper.

---

### Meta-Review · Area_Chair_YVP6 · 2024-12-11

**Metareview:**

A. Scientific Claims and Findings:

The paper compares the effectiveness of Transformer and LSTM architectures for financial forecasting using high-frequency trading data. The authors introduce a new LSTM-based model called DLSTM and a finance-specific Transformer. Their results indicate that Transformers have a slight advantage in absolute price predictions, but LSTMs are more reliable overall.

B. Strengths:

The paper addresses a relevant and significant question in financial time series forecasting.
The experimental setup is extensive and provides substantial data.
The paper is well-written and easy to follow.

C. Weaknesses:

Lack of code and detailed implementation information.
Limited novelty of the proposed approach.
The decomposition strategy is only applied to the LSTM model.
Several state-of-the-art Transformer-based models are not included in the comparison.
The statement about Transformers and LSTMs requires further investigation.
Limited dataset diversity.
Lack of ablation studies.
Limited discussion on model interpretability.

D. Reasons for Rejection:

The paper has several weaknesses, including limited novelty, lack of detailed information, and a limited comparison with state-of-the-art models. These issues raise concerns about the reproducibility and comprehensiveness of the research. Additionally, the paper lacks a deeper analysis of the observed differences between Transformer and LSTM models, which limits the interpretability of the results.

**Additional Comments On Reviewer Discussion:**

During the rebuttal period, the reviewers raised several concerns, including the lack of code, limited novelty, and the need for a more comprehensive comparison with state-of-the-art models. The authors responded to these concerns by stating that they will provide the code and include additional models in the revised manuscript. They also addressed the issue of novelty by emphasizing the unique contributions of their work, particularly in how DLSTM combines LSTM with time series decomposition.

Despite the authors' responses, the reviewers were not fully satisfied and decided to maintain their scores. The reviewers felt that the paper still required further improvements and that the authors' responses did not fully address their concerns.

In my final decision, I weighed each point raised by the reviewers and considered the authors' responses. I ultimately decided to reject the paper because I felt that the weaknesses outweighed the strengths. The limited novelty, lack of detailed information, and limited comparison with state-of-the-art models were major concerns that were not fully addressed during the rebuttal period.

---

### Decision · Program_Chairs · 2025-01-22

Reject